# Soil carbon in the world's tidal marshes

Tania L. Maxwell [1,31] ✉, Mark D. Spalding [1,2], Daniel A. Friess[3], Nicholas J. Murray [4], Kerrylee Rogers [5], Andre S. Rovai [6,7], Lindsey S. Smart[8,9], Lukas Weilguny [10], Maria Fernanda Adame [11], Janine B. Adams [12,13], William E. N. Austin[14,15], Margareth S. Copertino[16,17], Grace M. Cott[18], Micheli Duarte de Paula Costa [19], James R. Holmquist [20], Cai J. T. Ladd [21,22], Catherine E. Lovelock [23], Marvin Ludwig [24], Monica M. Moritsch[25], Alejandro Navarro[4], Jacqueline L. Raw [13,26], Ana-Carolina Ruiz-Fernández [27], Oscar Serrano[28], Craig Smeaton [14], Marijn Van de Broek[29], Lisamarie Windham-Myers[30], Emily Landis[8] & Thomas A. Worthington [1] ✉

Tidal marshes are threatened coastal ecosystems known for their capacity to store large amounts of carbon in their water-logged soils. Accurate quantification and mapping of global tidal marshes soil organic carbon (SOC) stocks is of considerable value to conservation efforts. Here, we used training data from 3710 unique locations, landscape-level environmental drivers and a global tidal marsh extent map to produce a global, spatially explicit map of SOC storage in tidal marshes at 30 m resolution. Here we show the total global SOC stock to 1 m to be 1.44 Pg C, with a third of this value stored in the United States of America. On average, SOC in tidal marshes' 0–30 and 30–100 cm soil layers are estimated at 83.1 Mg C ha$^{-1}$ (average predicted error 44.8 Mg C ha$^{-1}$) and 185.3 Mg C ha$^{-1}$ (average predicted error 105.7 Mg C ha$^{-1}$), respectively.

Tidal marshes, like other blue carbon[1,2] ecosystems (BCEs: mangroves, seagrasses and tidal freshwater emergent and forested wetlands), are a global soil organic carbon (SOC) hotspot owing to high rates of autochthonous and allochthonous organic matter deposition and slow decomposition in temporarily or permanently waterlogged soils. In addition to securing this carbon (C) over millennia, tidal marshes are also considered to be one of the most effective ecosystems for C accumulation[3]. Tidal marsh soils are capable of accreting vertically with sea level rise with inputs from allochthonous and autochthonous sources, thus limitations to C accumulation are far less likely to occur in marshes compared to terrestrial ecosystems, providing potential for continuous climate change mitigation benefits.

Tidal marshes were estimated to cover an area of 52,880 km$^2$ in 2020, distributed across 120 countries and territories[4], however, this is only a fraction of the prior extent. It is likely that over 50% of global tidal marsh habitat has been lost since 1800[5], with modern annual loss rates of between 0.2% and 2%[6–8] resulting from global warming, sea-level rise, and anthropogenic activities such as agricultural or urban expansion, and human engineering of coastal floodplains and river

systems[9]. Owing to the many ecosystem services tidal marshes provide, as well as their role in climate change mitigation, there is an increasing need to conserve and restore tidal marsh habitats globally[10,11].

As part of effectively managing tidal marshes and quantifying their climate mitigation potential, there is a need to first understand the quantities of C stored within their soils[12]. Though we have information on SOC in tidal marshes at local to regional scales in areas like the conterminous United States[13], Great Britain[14] and Australia[15], we currently lack a global scale analysis supported by extensive field-based observations and scaled up beyond temperate marshes[16]. Without this information, the scientific community and practitioners have to rely on global averages that are not ecosystem-specific[17] and that are based on data mainly from temperate regions[18], or they must collect resource-intensive in-situ field measurements.

Here, we present a global spatial model of SOC stock in tidal marshes and sampling bias-associated uncertainties. We did this by coupling a global tidal marsh extent map[4] with a global dataset containing 3710 measurements of tidal marsh soil properties and C

content[19,20], and using a machine-learning approach including environmental covariates identified by expert elicitation as potential drivers of soil C density. To align with IPCC and other guidance[21], we present spatially explicit SOC stocks to 1 m depths. This depth profile is used to represent soil more susceptible to emissions linked to disturbances. We also provide estimates for both the 0–30 cm and 30–100 cm soil layers, to align with both the numerous SOC studies that are confined to the upper layers and to match the conventional 30 cm depth for terrestrial soil C stock estimates.

## Results and discussion
### Global distribution of tidal marsh SOC
Our spatially explicit model predicts 1.44 Pg C in the top metre of tidal marsh soils globally. This estimate incorporates the spatial variability in tidal marsh SOC more adequately than previous studies, given that the model used training data from 3710 unique locations[19,20] and hypothesis-driven landscape-level drivers (Supplementary Table 1), while previous estimates have relied on averaged values from a smaller subset of data[22,23]. The data used to train the model are representative of most of the environmental conditions found in tidal marshes across the world (Supplementary Fig. 1), although representation is more limited from areas with different rates of Holocene relative sea-level rise (Supplementary Fig. 1h), certain coastal morphologies (Supplementary Fig. 1i), lower minimum temperatures (Supplementary Fig. 1k) and lower potential evapotranspiration (PET) rates (Supplementary Fig. 1n). Whilst our training dataset is extensive, there is also a bias in the geographic coverage of the training data, with over 85% from the USA, UK and Australia (Supplementary Fig. 2). Given the lack of data from the Arctic and the tropics, predictions from those regions are less certain and these are identified as locations for future assessments (see "Locations for priority sampling"). To account for these limitations in the training data, our model used an area of applicability (AOA)

approach[24] which identifies predictions where the environmental covariates are highly dissimilar to the environmental envelope captured by the training data. Due to the high expected error associated with the predictions outside the AOA, they were removed from our final SOC maps and statistics.

Previous global tidal marsh C stock estimates have taken a wide range of values. With lower values such as the $0.43 \pm 0.03$ Pg C estimated in the top 0.5 m from a dataset based mostly on North American tidal marshes[18], continental SOC averages to 1 m multiplied by extent estimates (1.41–2.44 Pg)[25], or ranging between 0.86 and 1.35 Pg C[16] estimated to a depth of 1 m from the SoilGrids map, a global machine-learning map from agricultural soils and terrestrial ecosystems data[17]. Conversely, simple calculations based on an average SOC value, applied to an overestimated tidal marsh extent have indicated that the global stock could be as high as 6.5 Pg C[22]. Our prediction of total global SOC in tidal marshes is significantly lower than this upper estimate, with our model predicting a range of 0.87–1.62 Pg C. It should be noted that our global tidal marshes SOC stock estimate is conservative because the underlying global map of tidal marshes we used only extends to 60° N and there are considerable unmapped areas of tidal marsh in the Arctic[4]. In addition, the combination of high expected error of predictions resulted in many areas in the tropics being removed from the statistics (Figs. S3 and S4). These removals, coupled with our currently incomplete understanding of the full distribution of tidal marshes[4], suggest that carbon stocks could also be underestimated in the tropics.

The magnitude of regional and national C stocks is strongly driven by tidal marsh area. Thus, a high proportion of the total global marsh C was located in the Temperate Northern Atlantic (Fig. 1a), which holds almost half (45%) of the global tidal marsh extent[4]. Countries with the highest predicted total SOC in tidal marshes (the U.S., Canada, and Russia, followed by Argentina, Australia, and Mexico—Fig. 1b and

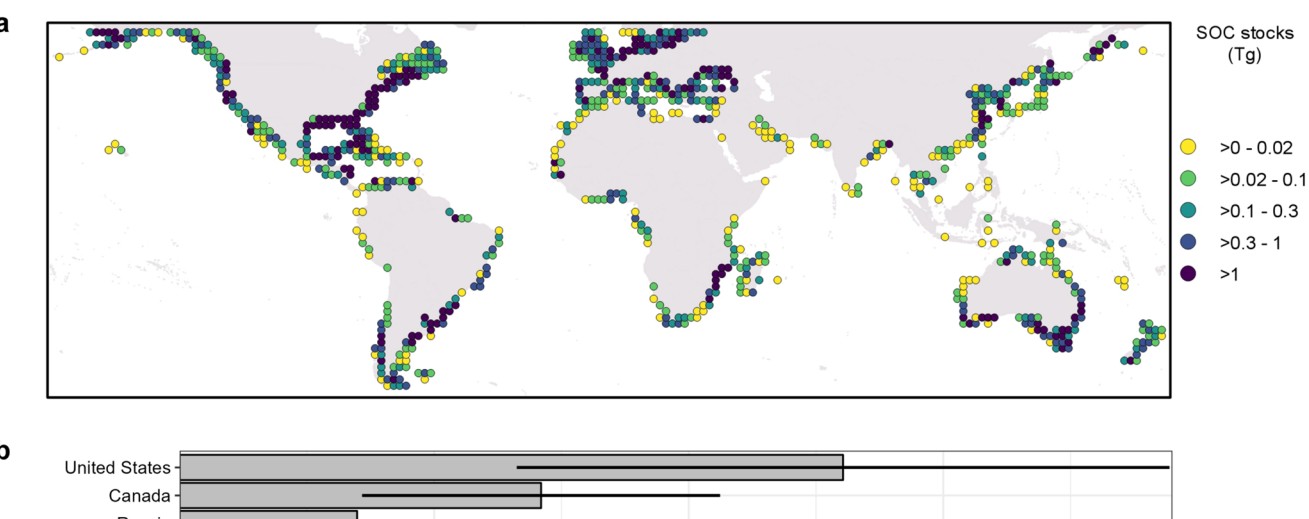

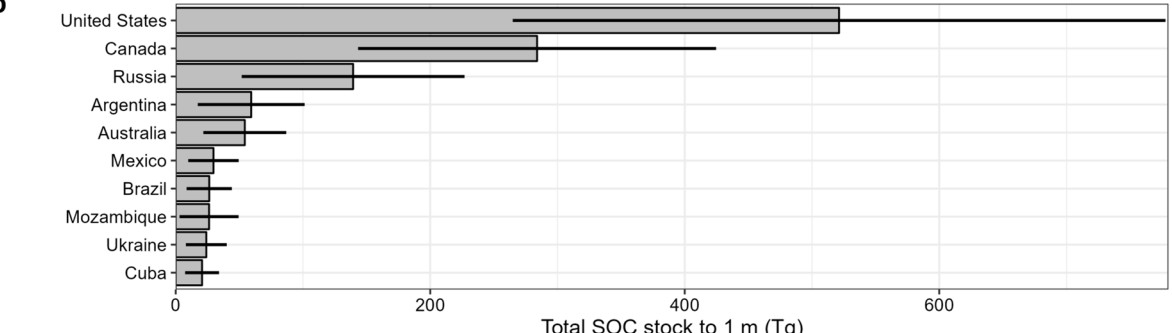

**Fig. 1 | Total SOC stocks (SOC, Teragrams (Tg)) in the top 1 m of tidal marshes.** **a** Aggregated per 2° cell, and (**b**) for the ten countries with the highest total SOC stock. Values refer to predicted SOC stocks after removing pixels outside the AOA, i.e. where we enabled the model to learn about the relationship between SOC stocks and the environmental drivers. Whiskers represent the expected model error.

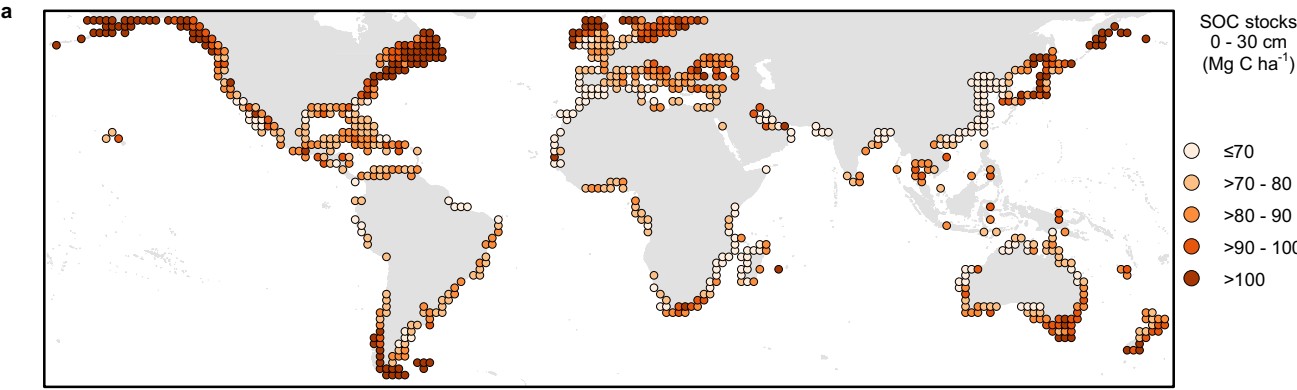

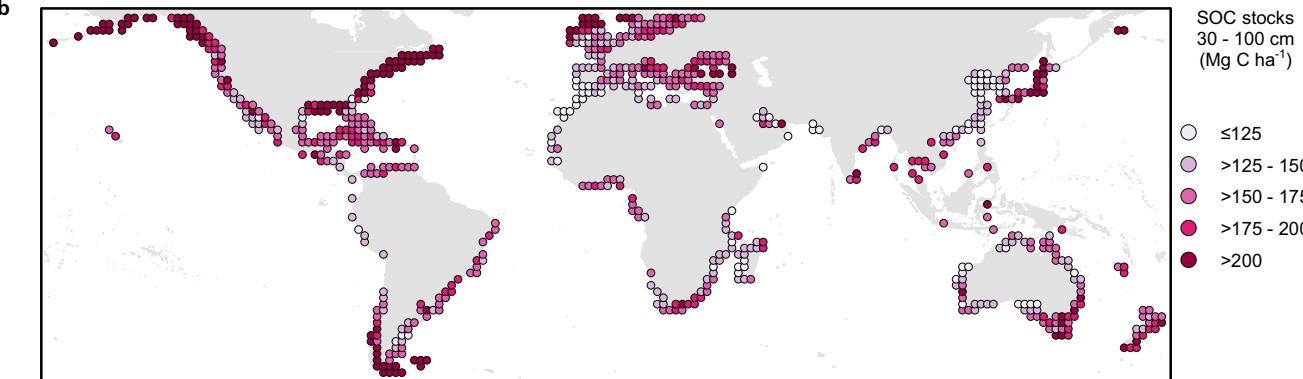

**Fig. 2 | Global distribution of tidal marsh SOC. a** For the 0–30 cm soil layer and (**b**) the 30–100 cm soil layer (aggregated per 2° cell). Values refer to predicted SOC per unit area (megagrams carbon per hectare (Mg C ha⁻¹)) after removing pixels outside the AOA, i.e. where we enabled the model to learn about the relationship between SOC and the environmental drivers. Cells with 0% of pixels within the AOA are not displayed. Because fewer training data points were available in the deeper soil layer, more pixels are outside the AOA and thus fewer cells are displayed in the lower panel. Initial predicted values and the proportion of pixels in each cell within the AOA are presented in Supplementary Figs. 5 and 6.

Supplementary Table 2) had both high C per unit area and large marsh extents[4]. Our country-level tidal marsh SOC stock estimates are consistent with those found in several regional studies. For example, stocks of 720 Tg C to 1 m were projected in the conterminous United States[13], which is comparable with our value of 520 Tg C. Similarly, 5.2 Tg C was estimated for the shallow (28 ± 16 cm) tidal marsh soils of Great Britain[26], which aligned with our prediction for the top 30 cm being 4.7 Tg C. Our results highlight the importance of geographical context when quantifying tidal marsh soil C stocks. For example, many marshes across Great Britain have relatively shallow soil profiles[26], and as such our estimate of 1 m would overestimate the national stock. Conversely, in settings with a longer history of relative sea-level rise or very high tidal ranges, marsh soil can reach depths exceeding 1 m[27,28], and thus we may underestimate total SOC stock. Not all of our findings are so well aligned with other studies. For example, we predict 19.3 Tg for China, while 57 Tg C was estimated in an earlier study (although this also included the contribution of mangroves and tidal flats)[29]. Such differences are likely to be driven by several factors, most strongly of which is the area of tidal marsh estimated for each country[26]. However, the availability of training data that accurately captures the variability of environmental conditions and the inclusion of finer-scale model predictors (e.g. data on tidal marsh plant communities) of C stocks will impact estimates.

We predicted that the average SOC per hectare in tidal marshes globally is approximately 83.1 Mg C ha⁻¹ in the 0–30 cm layer and 185.3 Mg C ha⁻¹ in the 30–100 cm layer (Fig. 2), with an average predicted error of 44.8 Mg C ha⁻¹ (Supplementary Fig. 3) and 105.7 Mg C ha⁻¹ (Supplementary Fig. 4), respectively. Our value for 1 m of 268 Mg C ha⁻¹ refines previous global estimates, such as

162 Mg C ha⁻¹ derived from local C data of unclear origin[22], and 317 Mg C ha⁻¹ averaged from an unspecified number of studies[23]. Our approach accounts for the spatial variability in C and is an improvement upon averages based on reported values alone. Further, our central estimate for tidal marsh soils is within the range of those predicted for mangrove soils (232–470 Mg C ha⁻¹)[30], confirming that these BCEs store significantly more SOC per unit area than many terrestrial ecosystems[31]. The average expected error associated with our predictions was reasonably consistent at the regional level (Supplementary Table 3, 0–30 cm layer: 43.0–52.5; 30–100 cm layer: 102.6–122.1); however, greater variation was more apparent at finer spatial scales (Supplementary Figs. 3 and 4).

Our analysis indicates larger SOC per hectare in higher latitudes, with northern areas of the Temperate Northern Atlantic and Pacific realms having particularly high values (Fig. 2). At the regional level, within both depth layers, temperate areas generally have slightly higher average SOC values than tropical regions (Supplementary Table 3), although there is high expected model error associated with each prediction (Supplementary Figs. 3 and 4). This finding goes against the hypothesis that higher temperatures are generally associated with higher SOC[32], due to the increase in productivity and growth of vegetation[33]. Instead, the lower soil temperature could limit SOC breakdown enhancing its storage potential[33], or temperature could be a weak driver at the global scale[34]. The large SOC predicted to 1 m in higher latitudes is influenced by limited training data and a low proportion of our predictions in the AOA (Fig. 3). In addition, the limited understanding of processes such as glacial isostatic adjustment and the impacts of relative sea level rise and how they influence C

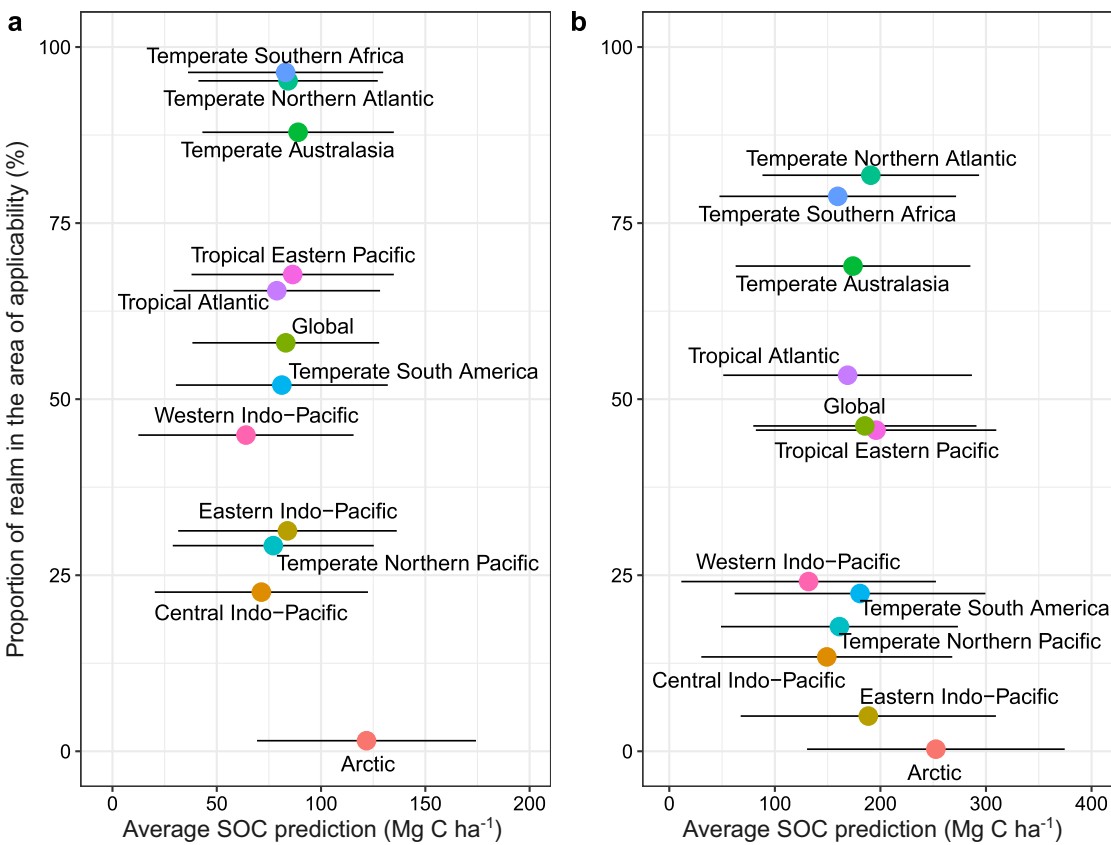

**Fig. 3 | Realm-level summary statistics of SOC.** In (**a**) the 0–30 cm soil layer and (**b**) the 30–100 cm soil layer. For each soil layer (0–30 cm and 30–100 cm), the x-axis shows the average final predicted SOC per unit area (megagrams carbon per hectare (Mg C ha⁻¹)), after masking out areas outside the AOA and the y-axis shows the proportion of the realm within the AOA, i.e. where we enabled the model to learn about the relationship between SOC and the environmental drivers. Whiskers represent the expected model error for each prediction. Colours are mapped to realms, which correspond to the biogeographical realms of the Marine Ecoregions of the World, and the global average.

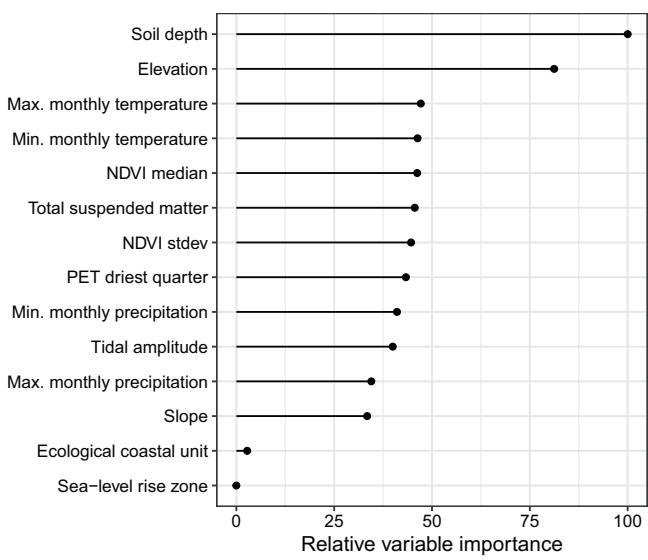

**Fig. 4 | Variable importance of the random forest model used to make the predictions of tidal marsh SOC stocks.** Max maximum, Min minimum, NDVI normalised difference vegetation index, stdev standard deviation, PET potential evapotranspiration. Importance was set to impurity in the model settings.

accumulation, decomposition and storage may profoundly alter estimates for the region, and remains an ongoing area of research.

### Drivers of SOC in tidal marshes

Overall, our model performed well in describing variation in our SOC training data, with an $R^2$ of 0.59. We selected hypothesis-driven environmental drivers (Supplementary Table 1) to limit the number of covariates included, as well as a spatial cross-validation strategy to avoid issues of overfitting[35]. Soil depth was the most important driver of SOC density (Fig. 4). This is consistent with findings quantifying SOC in mangrove ecosystems[30,36]. Depth has a strong influence on SOC concentration and bulk density, the interaction of which results in relatively stable SOC density measurements across the soil profile[13,19]. Elevation was also an important variable, with higher SOC values predicted at lower elevations, which are generally characterised by higher sedimentation rates allowing more trapping of organic C[37], as well as more frequent inundation providing an opportunity for deposition of tidally-distributed sediments to settle on tidal marsh surfaces[38] and for plants to allocate resources to their roots[39], adding organic material to the soils. Maximum SOC storage occurs just above mean sea level where (i) tidal marsh vegetation can thrive and serve as a C source, (ii) vertical space (or accommodation space) remains available for accumulation of organic matter enriched sediments from autochthonous and allochthonous sources, and (iii) decomposition of SOC is hampered by the anaerobic conditions created by higher inundation frequencies[34]. Temperature has been highlighted as being strongly

correlated with C stocks in coastal wetlands[32,40]; however, within our model temperature (both maximum and minimum) had similar relative variable importance as many other covariates (Fig. 4). While our model training data does not sample the full temperature covariate space for minimum temperatures (Supplementary Fig. 1k), other research has suggested that climate may not be a significant predictor of C stocks[34], as increased production and decomposition may balance out at higher temperatures. While the variable importance of the model indicates that temperatures partially drive the patterns of SOC, there is uncertainty in this assessment as underlying interactions between variables are not captured.

Our model includes the best globally available environmental covariates used to predict SOC in tidal marshes. However, there are a number of broad-scale drivers identified as potentially important predictors of SOC density which were poorly represented in our model or for which there were no globally available data products. For example, sea-level rise history has been linked to C storage in marshes[34,41]. This relationship was not apparent using our modelling method, but that is likely driven by the relatively coarse-scale classification of sea-level rise history zones[42] used in the analysis. While we included the normalised difference vegetation index (NDVI) as a proxy for tidal marsh vegetation type and productivity (the source of SOC), data sources representing the distribution of dominant plant species, diversity, or species assemblages at a global scale would be an important covariate to develop[43], as C stocks can vary with species and plant community[14,44]. Additionally, a high-resolution map of the coastal typology of tidal marshes would help refine predictions as geomorphic settings (for example deltas, estuaries, lagoons, composite deltas and lagoons, as well as sediment conditions and geology) influence the SOC stock via the type and rate of sediment supply to the coastline, nutrient loading/limitation, and organic matter diagenesis[40]. A global map of shoreline morphology would also better represent the accommodation space available for C storage[34,45], rather than using coastal typology. Finally, our analysis provides a static estimate of tidal marsh SOC stocks driven by environmental covariates. However, it is well established that natural and anthropogenic drivers can impact tidal marsh persistence and condition, and as such their SOC stocks[8]. Anthropogenic disturbances could both deplete C storage (e.g. from erosion or direct habitat removal, although such impacts would remove them from our map and model) or increase C storage (e.g.

from improved productivity due to nutrient additions) beyond what would be predicted by our model.

## Locations for priority sampling

While our global tidal marsh SOC model is underpinned by an extensive training dataset[19,20] of 42,741 observations from 3710 unique locations (Supplementary Fig. 1), the applicability of the model output is reduced in some regions where data is scarce or inexistent. Over 85% of the training locations are from the USA ($n = 2005$), the UK ($n = 944$) or Australia ($n = 284$). By implementing an AOA method[24] we restrict predictions to locations where the relationship between the training data and environmental covariates is meaningful. Due to the sparsity of training data from some locations, areas where our model predictions are robust (the AOA) represent 58% of mapped marshes globally for the 0–30 cm layer and 46.2% for the 30–100 cm layer.

The spatial applicability of our model varies between regions, with high AOAs (>85%) for many temperate regions, but very low applicability for the Arctic and much of the tropics (Fig. 3 and Table S3). By understanding this interplay between the predictions and their uncertainty, our analysis identifies priority areas for sampling to better parameterise future models (Fig. 5). For example, our analysis predicts high SOC across the high Arctic; however, this region is characterised by limited training data and thus high per pixel expected error. In our final analysis, these areas are outside the AOA, and therefore there are significant uncertainties when estimating SOC in this region. Many temperate areas were also predicted to have high SOC, but here predictions were underpinned by extensive field measurements and thus subject to greater certainty. Finally, many tropical and subtropical areas of tidal marsh were predicted to have low SOC. Tidal marshes in these regions, where mangroves are more likely to occur, are poorly characterised[46], and our predictions in these regions are limited by our knowledge of their structure and extent. SOC in tropical tidal marshes has been shown to be highly variable[47] and as such our model predictions will be sensitive to the limited training data available. Our analysis suggests that the average SOC per unit area is lower (Table S3, 0–30 cm layer: 5–23%; 30–100 cm layer: 9–29%) for three out of the five tropical regions compared to the global average; a result supported by the currently available studies[48].

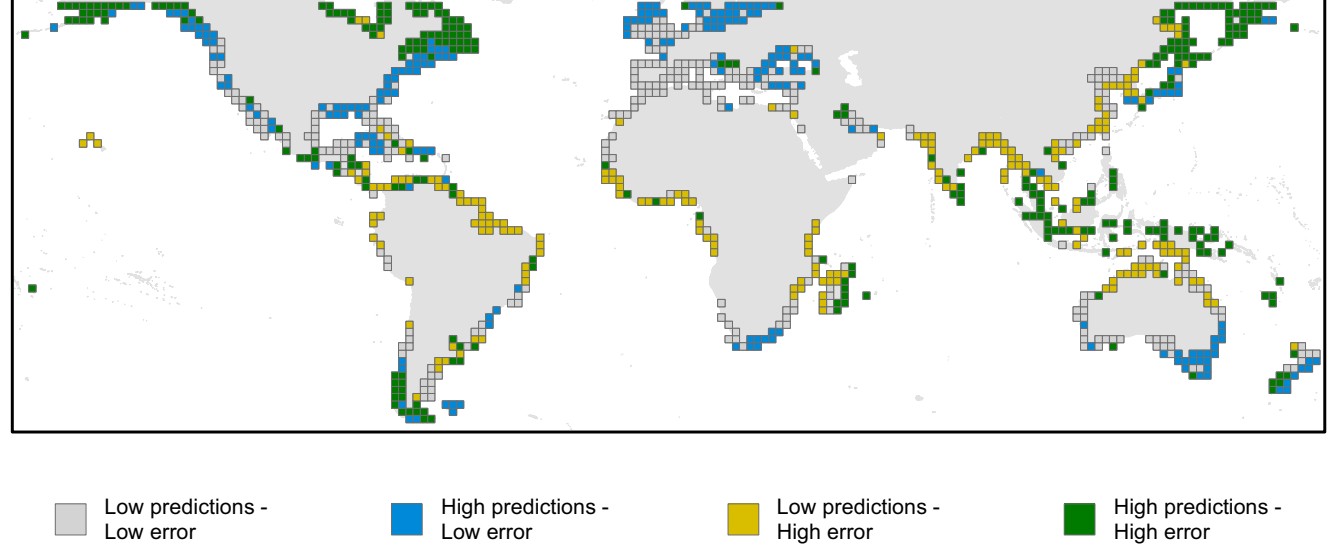

| Low predictions - Low error | High predictions - Low error | Low predictions - High error | High predictions - High error |

**Fig. 5 | Bivariate plot showing predicted SOC stocks per unit area and expected error.** Values are for the initial model predictions and expected model error (i.e. not masked by the AOA), aggregated per 2° cell. The plot shows locations with low predictions and low error (grey), high predictions and low error (blue), low predictions and high error (yellow), and high predictions with high error (green).

Our research highlights two key pathways for future work, firstly greater SOC field measurements from the extensive areas of Arctic tidal marsh to better quantify potential stocks in those regions, areas which are significantly threatened by climate change[49]. Secondly, we require far greater understanding of tidal marshes within the marsh-mangrove ecotone in the tropics to characterise their role as a key BCE. Tidal marshes in tropical regions are an important component of the coastal seascape, yet our understanding of both their extent and SOC stocks is limited by available data. Within our model there was variation in the predicted SOC per unit area across tropical realms (Table S3), but the majority of predictions were associated with high expected error (Fig. 5). The mechanisms behind the variability in SOC are unknown but may relate to variation in the community composition and productivity of tropical tidal marshes.

The uncertainties in our model also have a depth component, with consistently reduced AOAs in the 30–100 cm layer compared to the 0–30 cm layer (Fig. 3). This reduction is driven by the smaller number of data points in this deeper layer (35% of all data). This disparity in data coverage over soil depth could be addressed by more consistent sampling to 1 m across studies; however, not all marshes have an organic matter layer that exceeds 30 cm depth, as is often the case in the northern Pacific coast[50] and Great Britain[26]. These shallow, organic-rich tidal marsh soils formed more recently on top of tidal flats - thus the deeper, minerogenic layers do not hold equivalent levels of SOC found in the deeper organic-rich tidal marsh soils of many other locations. These findings indicate that estimating SOC stocks to 1 m following IPCC and other guidelines may not be appropriate in all cases, and as such we advocate providing SOC estimates across both the 0–30 cm and 30–100 cm layers.

## Outlooks and policy implications

Tidal marshes hold a substantial stock of SOC, and our globally consistent map improves knowledge of climate regulation services while practically supporting the inclusion of tidal marsh blue carbon in national inventories that could be used as a baseline for C accounting. Such data could catalyse coastal ecosystem conservation and restoration efforts. Disentangling sources of error from the available training data allows a greater understanding of prediction uncertainty, while our spatially explicit analyses of expected model error clearly highlight locations where future work is urgently needed. These areas include the Arctic and the tropics, and this work should include targeted field data collection of tidal marsh extent, plant community composition and C stocks. Our dataset enables quantification of potential losses in SOC from land use change or conversely benefits from conservation and restoration. International interest in BCEs is high, strongly driven by the calls for climate action under the United Nations Framework Convention on Climate Change (UNFCCC) and the UN Sustainable Development Goals. The very high C content and high sequestration rates of tidal marshes mean they can contribute to growing efforts to mitigate climate change by avoiding future losses, securing current stocks and also through ecosystem restoration[11,25]. A growing number of countries are including such ecosystems in their national commitments to climate change mitigation as part of their Nationally Determined Contributions, where our study can fill in significant gaps where data is scarce or is not available.

## Methods

### Tidal marsh soil carbon training data

The global database was compiled from two main sources: the recently produced global tidal Marsh Soil Organic Carbon (MarSOC) Dataset[19,51] (https://zenodo.org/records/8414110) and from the Coastal Carbon Research Coordination network[20,52] (https://ccrcn.shinyapps.io/CoastalCarbonAtlas/). The data were filtered to those sites located between 60° N to 60° S due to the environmental covariate data being limited to this region, and within a coastal zone data mask[53]. This dataset contains 42,741 data points from 22 countries and 3710 unique locations[27,29,32,37,48,54–247].

### Spatial modelling of SOC

We used a 3D approach to model organic carbon density (OCD) to maximise the applicability of the collected data and reduce the need to make assumptions about OCD trends along the soil profile[248]. We thus modelled OCD as a function of depth ($d$), and a series of environmental covariate layers ($X_n$):

$$OCD(xyd) = d + X_1(x, y) + X_2(x, y) + \dots X_n(x, y) \tag{1}$$

where $xyd$ are the 3D coordinates in decimal degrees of latitude and longitude, and soil depth (measured at the centre of a sampled soil layer). The resulting spatial prediction model can then be used to predict OCD at standard depths of 0 cm, 30 cm, and 100 cm, so that the SOC for the 0–30 cm and 30–100 cm soil layer can be calculated using their respective layer thicknesses[248]:

$$SOC\ stock_{0-30cm} = (OCD_{0cm} + OCD_{30cm}) / 2 * 30cm \tag{2}$$

$$SOC\ stock_{30-100cm} = (OCD_{30cm} + OCD_{100cm}) / 2 * 70cm \tag{3}$$

Thirteen environmental covariates were used based on hypothesised landscape-level drivers of SOC in tidal marshes (Table S1). Drivers were selected by the authors whose expertise includes regional and field-based knowledge as well as considerable prior knowledge of global-scale modelling. This was an iterative process with group discussion and feedback. It was also, however, constrained by data availability. We included covariates representative of potential ecological, environmental, and geomorphological drivers.

We used the NDVI as a proxy for distinguishing vegetation type and the source of SOC. NDVI has been shown to be able to discriminate between broad classes of tidal marsh vegetation[249,250]. To calculate the NDVI metrics (median and standard deviation), we used Landsat 8 bands from 2014 to 2021 available at 30 m resolution, courtesy of the U.S. Geological Survey, and image processing code from Murray et al.[53,251] in Google Earth Engine[252].

Elevation and slope data were included, as they can be a proxy of soil age and composition, as well as vegetation structure in marshes[253]. Additionally, higher SOC stocks may be associated with shallower slopes, due to a lower risk of erosion compared to steeper slopes[254]. We used the Copernicus Digital Elevation Model GLO-30 dataset, which was developed from the TanDEM-X mission between 2011 and 2015. The product is a global dataset of elevation at 30 m resolution and has an absolute vertical accuracy of less than four metres. The slope was then derived using the ee.Terrain.slope() function in Google Earth Engine[252]. The dataset can be found here: https://spacedata.copernicus.eu/collections/copernicus-digital-elevation-model.

Tidal amplitude can influence the stability and resilience of marshes[255], as well as the accommodation space[45]. We used the FES2014 Tide Model M2, which has a resolution of ~7 km, available here at https://datastore.cls.fr/catalogues/fes2014-tide-model/.

Higher SOC stocks further from estuaries can be explained by the signature of past sea level rise[34]. We used the five broad classes originally presented in Clark et al.[42].

The total suspended matter (TSM) was retrieved from the GlobColour project, which processed the TSM data from MERIS imagery collected by the Envisat European Space Agency satellite. Monthly values at 4 km resolution were retrieved from the period 2003–2011, as these were the full years available, which were averaged to generate one TSM layer. The data can be found here: https://hermes.acri.fr/.

Each coastal setting (deltas, estuaries, lagoons, composite deltas and lagoons, bedrock, and carbonate) has an environmental signature

that controls the SOC stock via the type and rate of sediment supply to the coastline, nutrient loading/limitation, and organic matter diagenesis[40]. We used the groups of coastlines from the Ecological Coastal Units[256], which were generated by clustering the 4 million coastal line segments based on ten variables (two land, five ocean, and three coastline variables). We rasterized the 1 km shoreline segments, which were available from: https://www.arcgis.com/home/item.html?id=54df078334954c5ea6d5e1c34eda2c87.

Higher temperatures generally increase the productivity and growth of vegetation[33], and are associated with higher SOC stocks[32]. Higher rainfall is generally associated with higher SOC by increasing the freshwater runoff and thus potentially higher deposition of allochthonous organic matter[254]. Both minimum and maximum monthly average values of temperature and precipitation were chosen rather than mean annual values[40], as they portray environmental thresholds that may have a stronger effect on SOC stocks by constraining ecosystem functionality[257], which regulates both production and decomposition rates. We used data from WorldClim BIO variables[258] at 927.67 m resolution, available from: https://www.worldclim.org/data/bioclim.html.

PET is the amount of plant evaporation that would occur if there was a sufficient water source in the surrounding soil. This measure has been found to explain different ecophysiological processes in mangroves[259], and may show a tradeoff in extreme climates. We used the average ENVIREM mean monthly PET of the driest quarter[260], as this can represent the more extreme cases, when a prolonged dry period can have a negative effect on plant productivity. The dataset can be found here: https://envirem.github.io/#downloads.

Most covariate layers (tidal amplitude, TSM, ECUs, temperature, precipitation, PET) were land or ocean products that needed to be extrapolated to each pixel containing tidal marsh. This was done by calculating the average of neighbouring pixels using a circle-shaped boolean kernel in Google Earth Engine[252], with the functions ee.Image.reduceNeighborhood(), either ee.Reducer.mean() or ee.Reducer.mode() for categorical variables, and ee.Kernel.circle(1, 'pixels'). To get all covariate map layers to the same 30 m resolution as the DEM and the NDVI layers, we re-sampled and re-projected coarser resolution data to a unified pixel grid (World Geodetic System 1984, EPSG:4326) using bilinear interpolation. Collinearity between the continuous variables was visualised using the corrplot package (version 0.92)[261], and identified generally low correlations between the variables (Supplementary Fig. 7), and below the threshold of $|r| > 0.7$[262].

We visualised how well our training data captured the variability of the environmental covariates across the global tidal marsh extent, and as such, how potentially biased the environmental covariate data that was used to train the model was. To do this we sampled the covariate values for 10,000 points drawn randomly from across the global tidal marsh extent and compared these to the covariate values for the training locations (Supplementary Fig. 1).

## Model training

We used the random forest model implemented in the ranger package (0.15.1)[263] and the caret framework (6.0-94)[264] within R[265]. Due to the limited data in many regions of the world, we used the entire dataset for model training. To test model accuracy, $R^2$ and Root Mean Square Error (RMSE) values were calculated from all pairs of observed and predicted response values when held back from model training using the cross-validation method.

We used resampling-based cross-validation to provide an estimate of the predictive performance of the random forest model[266]. The choice of a cross-validation strategy is key because it determines the estimation of the model performance as well as the variable importance. Our training data is clustered due to the nature of the available measurements (i.e. large amounts of data in the USA, Australia, and the U.K.), which means that random cross-validation (CV) would only

indicate the ability of the model to predict within these clusters[35,267,268]. A spatial cross-validation strategy, in which spatial units are held back for validation[268,269], assesses the ability of the model to predict beyond the clusters, which is in line with the purpose of the model to predict into spaces that lack training data[269–271]. We used the k-fold nearest neighbour distance matching (k-NNDM) cross-validation presented by Linnenbrink et al.[272], implemented within the CAST package (0.8.1)[273], which is a variant of the leave-one-out NNDM cross-validation with reduced computation time compared to the method developed by Milà et al.[270].

This k-NNDM method creates folds such that the geographic distance between sample points of different folds approximates the distance between the training samples and the prediction locations. To create the prediction locations, 5000 points were randomly selected from the global tidal marsh extent map[4]. We used $k = 5$ folds (Supplementary Fig. 8), so that the training data were clustered geographically. When the model separates the training data into training and testing at the cross-validation phase, each fold serves as a testing set while the others are used for training. By comparing the geographic distance between folds of our k-NNDM CV to those between random folds, we can see that our method better resembles the distance from prediction locations to training samples (Supplementary Fig. 9).

We implemented a model tuning step, testing 100 possible models by varying the number of variables to consider at each split (1, 2, 3, 4, or 5), the minimum node size (1, 2, 3, 5, or 10), and the number of trees (100, 200, 300, 400, or 500). Between these, the RMSE varied very little, as expected for random forest models[274]. We thus set mtry to 3, the minimum node size to 5, and the number of trees to 300.

Within the final random forest, variable importance was set to "impurity" within the ranger package, corresponding to the Gini index for classification. This was used to identify the relative importance of the environmental covariates to the SOC predictions.

## Predictions

To align with the highest resolution covariate variable, the 10 m tidal marsh data was exported at 30 m resolution using Google Earth Engine[252]. The predictions were made for every 30 m pixel identified as a tidal marsh in 2020[4]. This recent extent map was derived from earth observation data and estimates 52,880 km² of tidal marshes between 60° N and 60° S. As described in the spatial modelling of the "SOC" section, the model predicted SOC density at 0 cm and 30 cm, which were then averaged and multiplied by 30 cm (the layer thickness) and by 100 to get a SOC stock in Mg per hectare. This was also undertaken for the 30–100 cm soil layer, using the SOC density values at 30 cm and 100 cm. Thus, each 30 m pixel has a predicted SOC stock (Mg ha⁻¹) for the 0–30 cm and 30–100 cm soil layers.

## Pixel-wise accuracy estimation

To estimate how different the prediction areas were from areas on which the model was trained, we first calculated a dissimilarity index (DI). In brief, this is calculated by dividing the minimum distance to the nearest training data point in a multidimensional predictor space that has been scaled and weighted by variable importance, by the average of the distances in the training data[24]. The DI is calculated based on data points that do not occur in the same cross-validation fold, thus keeping in mind the cross-validation nature of the model.

Then, we used the relationship between the DI and the prediction performance (i.e. the final model RMSE) to produce a spatially continuous estimation of the expected error associated with each SOC prediction (DItoErrormetric() in the developer's CAST commit version from August 2023)[273]. This uses shape constrained additive models[275] to model the relationship between the DI and the RMSE. This model can then be applied to the DI of every tidal marsh pixel to produce a spatially continuous map of the estimated accuracy of the predictions. We calculated the expected model error similarly as we did for the

predictions using the spatial modelling approach, and to get an expected error in the same units as the predictions.

The analysis workflow (model training, predictions, error, and AOA) was completed using Snakemake[276]. We calculated average prediction and expected error values for eleven out of the twelve biogeographical realms of the Marine Ecoregions of the World[277] (Fig. 3 and Table S3). There was no tidal marsh extent predicted in the twelfth realm, Southern Ocean. We also calculated averages per country using the union of the ESRI Country shapefile and the Exclusive Economic Zones from the Flanders Marine Institute[278].

## AOA
Although we can apply the Random Forest model to all tidal marshes globally because of the availability and preparation of covariate data for all marshes, these predictions can often be extrapolated and rendered meaningless when predictor values are too different compared to the training data[271]. To ensure our predictions were bounded with the environmental envelope of our training data, we implemented the AOA methodology, introduced by Meyer and Pebesma [24], to mask out areas where the model was not able to learn about the relationship between the predictors and the response (here, SOC density). We specifically excluded areas with a different covariate space where predictions of carbon stocks would be uncertain because of a lack of training and validation data. The threshold for determining the AOA was based on the outlier-removed maximum DI of the training data, i.e. data larger than the 75th percentile plus 1.5 times the interquartile range of the DI values of the cross-validated training data. The calculation of the pixel-level DI and the AOA were generated using the aoa() function, both available in the CAST package (0.8.1)[273]. Then, the predictor space which is greater than the AOA threshold is considered outside the AOA, and thus is masked from our predictions. For each predicted soil layer (i.e. 0–30 cm and 30–100 cm), the AOA was calculated at the upper and lower depths and then averaged. Locations considered inside the AOA (Supplementary Figs. 5b and 6b) are those where the averaged AOA is equal to 1 (i.e. AOA values of 0 or 0.5 are considered outside the AOA).

## Data availability
The training data, tidal marsh extent and environmental covariate data used in this study are publicly available (linked in the Methods). The soil organic carbon predictions, estimated model errors, and area of applicability layer for both the 0–30 cm and 30–100 cm layers are available on Zenodo (https://doi.org/10.5281/zenodo.10940066).

## Code availability
All code used in this study is available on Github[279]: https://github.com/Tania-Maxwell/global-marshC-map/.

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

## Acknowledgements

We thank Daniele Baisero, Thomas Ball, and Alison Eyres for methodological help. This project benefited from funding from the Bezos Earth Fund and other donors supporting the Nature Conservancy (T.A.W., E.L., and M.D.S.). LH Pérez-Bernal provided assistance in the geochemical analysis of sediment cores from Mexico. This work was performed using resources provided by the Cambridge Service for Data-Driven Discovery (CSD3) operated by the University of Cambridge Research Computing Service (www.csd3.cam.ac.uk), provided by Dell EMC and Intel using Tier-2 funding from the Engineering and Physical Sciences Research Council (capital grant EP/T022159/1), and DiRAC funding from the Science and Technology Facilities Council (www.dirac.ac.uk). Any use of trade, firm, or product names is for descriptive purposes only and does not imply endorsement by the U.S. Government.

## Author contributions

T.A.W., M.D.S., E.L., and T.L.M. designed the research. T.L.M. and T.A.W. performed the analyses. M.F.A., J.A., W.E.N.A, M.S.C., M.D.d.P.C., G.C., D.F., J.H., C.J.T.L., C.E.L., M.M.M., J.R., K.R., A.C.R.-F., O.S., C.S., M.V.d.B., and L.W.-M. provided expert knowledge on tidal marsh soil carbon dynamics. C.J.T.L. M.L., N.M., A.N., A.R., and L.W. helped with the methodology. E.L. provided funding. M.D.S. L.S.S. and T.A.W. provided supervision. The manuscript was draughted by T.L.M. and T.A.W. with contributions from all co-authors.

## Competing interests

The authors declare no competing interests.

## Additional information

[1]Conservation Science Group, Department of Zoology, University of Cambridge, Cambridge, UK. [2]The Nature Conservancy, Siena, Italy. [3]Department of Earth and Environmental Sciences, Tulane University, New Orleans, LA, USA. [4]College of Science and Engineering, James Cook University, Townsville, QLD, Australia. [5]Environmental Futures, School of Science, University of Wollongong, Wollongong, NSW, Australia. [6]U.S. Army Engineer Research and Development Center, Vicksburg, MS, USA. [7]Department of Oceanography and Coastal Sciences, Louisiana State University, Baton Rouge, LA, USA. [8]The Nature Conservancy, Arlington, VA, USA. [9]Department of Forestry and Environmental Resources, NC State University, Raleigh, NC, USA. [10]European Molecular Biology Laboratory, European Bioinformatics Institute, Wellcome Genome Campus, Hinxton, UK. [11]Australian Rivers Institute, Coastal and Marine Research Centre, Griffith University, Nathan, QLD, Australia. [12]Department of Botany, Nelson Mandela University, Gqeberha, South Africa. [13]Institute for Coastal and Marine Research, Nelson Mandela University, Gqeberha, South Africa. [14]School of Geography and Sustainable Development, University of St Andrews, St Andrews, UK. [15]Scottish Association of Marine Science, Oban, UK. [16]Federal University of Rio Grande (FURG), Rio Grande, Brazil. [17]Brazilian

Network of Climate Change Studies—Rede CLIMA, Rio Grande, Brazil. [18]School of Biology and Environmental Science, University College Dublin, Belfield, Ireland. [19]Deakin Marine Research and Innovation Centre, School of Life and Environmental Sciences, Deakin University, Warrnambool, VIC, Australia. [20]Smithsonian Environmental Research Center, Edgewater, MD, USA. [21]Department of Geography, Swansea University, Swansea, UK. [22]School of Ocean Sciences, Bangor University, Menai Bridge, UK. [23]School of Environment, The University of Queensland, St Lucia, QLD, Australia. [24]Institute of Landscape Ecology, University of Münster, Münster, Germany. [25]University of California, Santa Cruz, Santa Cruz, CA, USA. [26]Anthesis South Africa, Cape Town, South Africa. [27]Instituto de Ciencias del Mar y Limnología, Universidad Nacional Autónoma de México, Unidad Académica Mazatlán, Mazatlán, Mexico. [28]Centro de Estudios Avanzados de Blanes (CEAB), Blanes, Spain. [29]Swiss Federal Institute of Technology (ETH Zürich), Zürich, Switzerland. [30]California Delta Stewardship Council, Sacramento, CA, USA. [31]Present address: Biodiversity, Ecology and Conservation Research Group, International Institute for Applied Systems Analysis (IIASA), Laxenburg, Austria. ✉e-mail: taniamaxwell7@gmail.com; taw52@cam.ac.uk

