## [Peer Review file · Nature Communications]

Soil carbon in the world's tidal marshes

Corresponding Author: Dr Thomas Worthington

Version 0:

Reviewer comments:

Reviewer #1

(Remarks to the Author)

Response to the Authors:

The manuscript "Soil carbon in the world's tidal marshes" presents results from a global soil organic carbon allocation study using large collaborative datasets and spatially explicit random forest modeling. The goal of this study is to quantify the SOC stored down to 1 meter in global tidal marshes, as well as investigate potential parameters as controls for SOC. The authors estimate a total of 1.44 Pg C stored globally at approximately 83.1 Mg ha⁻¹ and 185. Mg ha⁻¹ for the 0-30 cm and 0-100 cm respectively. The authors also determine soil depth to be the greatest indicator of SOC stocks, with elevation as second most important.

This study uses an extensive amount of data, and the model development and usage are carefully considered. While this study and methodology is a valuable and much needed analysis, I have some concerns and would like more clarification before I trust the reliability of this as a global analysis. In general, the methods, uncertainty analysis, and statistics need to be explained more thoroughly. As an analysis of temperate locations, this is a robust estimation. However, the lack of data in the tropics and Arctic, along with some lack of clarification within the methods, makes me question the reliability of the current model outside of the temperate locations (or locations with training data or similar to those with training data) to those locations that are significantly different.

For example, the dataset is global, but lacking many key regions. Certain environmental covariates, such as temperature, that have been considered important in other assessments are not as influential in this study. Is that because temperature is not as important as other parameters, or because the training/sampling data is overwhelming from only temperate regions and the tropics/ Arctic are excluded from the AOA (lines 486-501/ Fig S1 S2)? If cross-validation is the only way that the importance of the covariates is determined, shouldn't the AOA be the entire area in consideration?

The environmental covariates used were expertly chosen and have been commonly used and tested in SOC variability studies. While using these variables within the cross-validation portion of the model makes sense, the use of these parameters does not show any particularly novel results globally. Many of these parameters are just general proxies and include multiple different ecosystem variables. I do think that the authors missed an opportunity with this large of a dataset to show potential insight into global trends of SOC, but also potential trends in those regions that are currently lacking data.

In addition, the authors describe that currently the researchers have to rely on averages that are heavily biased towards temperate regions or not ecosystem specific but given the lack of data in other regions such as the tropics and the training data heavily in temperate regions, I'm not sure why this estimation is less biased. Did the authors test the bias? There needs to be a more extensive bias analysis for the authors to reliably say this study is less temperately biased than the others.

Individual Lines:

Line 85: Need to define SOC within the Main, not just the abstract.

Line 105-108: The authors need more support to be able to say this. See temperature example in summary.

Fig 3: Are any of these averages significantly different from one another? All the regions look very similar. It is strange that the Arctic has the highest average SOC.

Line 185: How are the whiskers describing the error for the region, yet all of the whiskers show the same length? These don't seem to match up with the potential for value error shown in Fig 5, S3, and S4.

Line 203-205: How was the importance of these variables tested? Was it only in the cross-validation? Was no collinearity

tested? The authors need to describe the methodology in much more detail. Environmental variables are notoriously collinear, yet these lines imply that the authors did not test for variable interactions.

Line 255: How much lower for tropics compared to global? The model does not show that much of a decrease between the averages or ranges.

Lines 488: How different is too different? From Figures S1 and S2, it appears that a significant portion of the tropics and Arctic are not included in the AOA. With so much error, what can truly be determined from these estimations? Besides the fact that these areas need more ground-truthing.

References:

Why do the references suddenly go in alphabetical order halfway through? This formatting may need to be checked.

Supplemental:

Table S1: Are these variables decided by literature or by expert opinion/discussion (Lines 331-334)? Otherwise, very helpful and informative table.

Table S2: Also, good table.

Figure S4: The color choices for plot a and c make it difficult to see any variation in the maps. Given these are the key error maps for the 30-100 meter, these should be as easy to see as Figure S3.

(Remarks on code availability)

Reviewer #2

(Remarks to the Author)

General Comments:

The manuscript presents a compelling methodology for quantifying the current carbon stocks in coastal wetlands globally. The methodology utilises a broad literature base for measured SOC at specific locations, a new global tidal marsh extent map and a machine-learning approach including environmental covariates (identified by a broad input from co-authors) as potential drivers of soil carbon density. The manuscript does not, however, attempt to discuss discrepancies with other methodologies for determining SOC stocks, for example, the manuscript describes the estimate for China to be 19 Tg, while a recent paper in GCB (Xia et al) estimates the stocks to exceed 50 Tg.

Table 4 describes the key drivers for SOC stocks using random forest methodology, however, it is not entirely clear what the NDVI was reporting- was this used as a proxy for vegetation density or vegetation type or both? Perhaps this needs clarification in the methods.

The manuscript provides valuable insights into regions that require empirical data to further train the models. In particular, the Arctic regions and tropical regions require further attention. It would be interesting to further understand why mangrove forests in the tropics give high variability in predictions.

The outlook and policy implication section is also compelling, based on data presented in the manuscript and a sound understanding of ecosystem restoration.

The model is trained with a robust range of environmental covariates. However, anthropogenic disturbance is not factored into model outputs, while it is generally well understood that degradation via erosion (as an example) can also severely impact SOC stocks. Perhaps these areas were excluded from the current assessment, but this may need some further explanation.

The manuscript is generally very well written and highly polished. There could perhaps be improved consistency in the use of abbreviations/ chemical symbols, in particular carbon and C are used interchangeably, and should be consolidated to C. Otherwise, the manuscript is well suited for Nature Communications and should achieve significant utilization by a broad audience.

Specific comments:

54 "used training data from 3,710 unique locations" rather, the model is trained on a globally distributed empirical dataset. This minor amendment may improve the readability for a non-expert in modelling approaches.

67 SOC

68 waterlogged and temporarily waterlogged soils?

70-72 The concept of C saturation in terrestrial systems is driven by edaphic controls (mainly), while this is less relevant in wetlands- where protection of C relies on protection of POM and MAOM by lower microbial oxidation. The increase in level of wetlands will rely heavily on allochthonous inputs to bring in the mineral matter. I think this sentence could use a little more explanation.

91 in-situ in italics?

118-120 The high variability in tropical regions- particularly mangrove forests, may also be underestimated. Perhaps this should be mentioned here.

323 not sure that SOC needs to be redefined here

366 Total does not require capitalization.

491 SOC

(Remarks on code availability)

As mentioned above, I am not a coding expert and hope that other reviewers will be able to complete a review of this aspect.

Version 1:

Reviewer comments:

Reviewer #1

(Remarks to the Author)

(Remarks on code availability)

Reviewer #2

(Remarks to the Author)

The authors have carefully addressed the comments from my review, and that of another reviewer, with additional clarification provided in the manuscript. The response to reviewers was thorough. It is my opinion that the manuscript is ready for publication.

(Remarks on code availability)

Response to Reviewers

Reviewer #1 (Remarks to the Author):

Response to the Authors:

We thank the reviewer for the insightful comments on the manuscript, we have addressed each in turn below.

The manuscript “Soil carbon in the world’s tidal marshes” presents results from a global soil organic carbon allocation study using large collaborative datasets and spatially explicit random forest modeling. The goal of this study is to quantify the SOC stored down to 1 meter in global tidal marshes, as well as investigate potential parameters as controls for SOC. The authors estimate a total of 1.44 Pg C stored globally at approximately 83.1 Mg ha⁻¹ and 185. Mg ha⁻¹ for the 0-30 cm and 0-100 cm respectively. The authors also determine soil depth to be the greatest indicator of SOC stocks, with elevation as second most important.

This study uses an extensive amount of data, and the model development and usage are carefully considered. While this study and methodology is a valuable and much needed analysis, I have some concerns and would like more clarification before I trust the reliability of this as a global analysis. In general, the methods, uncertainty analysis, and statistics need to be explained more thoroughly. As an analysis of temperate locations, this is a robust estimation. However, the lack of data in the tropics and Arctic, along with some lack of clarification within the methods, makes me question the reliability of the current model outside of the temperate locations (or locations with training data or similar to those with training data) to those locations that are significantly different.

We have added extra text to the methods to clarify the differences between the cross-validation and variable importance, and provided further explanation of the area of applicability (AOA). We have also added more information to Table S1 to better describe the variables.

We agree with the reviewer that there is a lack of training data from the Arctic and tropics; however, we believe our approach to explicitly remove those predictions outside the AOA to some extent mitigates those biases in the training data. In addition, while the training data is biased towards temperate areas, estimates of the distribution of tidal marshes suggests almost two-thirds of the global extent is within temperate regions.

To address the reviewers comments we have taken the following steps:

- 1) We have reworked the start of the paper to highlight both the limitations of the training data and also how the AOA approach tries to address these limitations.

“To account for these limitations in the training data, our model used an area of applicability (AOA) approach which identifies predictions where the environmental covariates are highly dissimilar to the environmental envelope captured by the training data. Due to the high expected error associated with the predictions outside the AOA, they were removed from our final SOC maps and statistics.”

- 2) We provide an analysis of the distribution of environmental covariates for the training and compare it to 10,000 randomly sampled points from across the global tidal marsh extent. This visualisation highlights that our training data is reasonably representative of

the environmental conditions found in tidal marshes across the world. However, we highlight those areas where the training data lacks coverage of certain portions of the environmental covariate space.

“The data used to train the model are representative of most of the environmental conditions found in tidal marshes across the world (Fig. S1), although representation is more limited from areas with different rates of Holocene relative sea-level rise (Fig. S1h), certain coastal morphologies (Fig. S1i), lower minimum temperatures (Fig. S1k) and lower potential evapotranspiration (PET) rates (Fig. S1n). Whilst our training dataset is extensive, there is also a bias in the geographic coverage of the training data, with over 85% from the U.S.A. U.K. and Australia (Fig. S2).”

- 3) In the text we highlight that predictions in certain areas such as the tropics and Arctic should be treated with caution.

“Given the lack of data from the Arctic and the tropics, predictions from those regions are less certain and these are identified as locations for future assessments (see Locations for priority sampling).”

- 4) In the text we have also highlighted the limited understanding of processes such as glacial isostatic adjustment and relative sea level rise operating in Arctic areas, and removed the explanation of potential drivers due to this uncertainty.

“In addition, the limited understanding of processes such as glacial isostatic adjustment and the impacts of relative sea level rise and how they influence C accumulation, decomposition and storage may profoundly alter estimates for the region, and remains an ongoing area of research”

For example, the dataset is global, but lacking many key regions. Certain environmental covariates, such as temperature, that have been considered important in other assessments are not as influential in this study. Is that because temperature is not as important as other parameters, or because the training/sampling data is overwhelming from only temperate regions and the tropics/ Arctic are excluded from the AOA (lines 486-501/ Fig S1 S2)? If cross-validation is the only way that the importance of the covariates is determined, shouldn't the AOA be the entire area in consideration?

Within the modelling framework cross-validation is used to assess the model's predictive performance. We have added a sentence to the methods to clarify this.

“We used resampling-based cross-validation to provide an estimate of the predictive performance of the random forest model.”

The importance of the covariates is assessed using the variable importance functions within the random forest model. We have added an expanded explanation of this in the methods section.

“Within the final random forest, variable importance was set to “impurity” within the ranger package, corresponding to the Gini index for classification. This was used to identify the relative importance of the environmental covariates to the SOC predictions.”

The area of applicability (AOA) approach is used to identify the predictions of the model that are not too dissimilar to the covariate space of the predictors used to train the model. We have

added more explanation in the methods section. These regions are excluded methodologically and systematically by the design of our method.

“To ensure our predictions were bounded with the environmental envelope of our training data, we implemented the area of applicability (AOA) methodology, introduced by Meyer and Pebesma 2021²⁴, to mask out areas where the model was not able to learn about the relationship between the predictors and the response (here, SOC density). We specifically excluded areas with a different covariate space where predictions of carbon stocks would be uncertain because of a lack of training and validation data.”

And highlight this in the first paragraph of the results section.

“To account for these limitations in the training data, our model used an area of applicability (AOA) approach which identifies predictions where the environmental covariates are highly dissimilar to the environmental envelope captured by the training data. Due to the high expected error associated with the predictions outside the AOA, they were removed from our final SOC maps and statistics.”

In terms of temperature we highlight its role in determining C stocks in the following sentences

“This finding goes against the hypothesis that higher temperatures are generally associated with higher SOC³², due to the increase of productivity and growth of vegetation³³. Instead, the lower soil temperature could limit SOC breakdown enhancing its storage potential³³, or temperature could be a weak driver at the global scale³⁴.”

And have added a further explanation in the ‘Drivers of soil organic carbon in tidal marshes’ section.

“Temperature has been highlighted as being strongly correlated with C stocks in coastal wetlands^{32,40}; however, within our model temperature (both maximum and minimum) had similar relative variable importance as many other covariates (Fig. 4). While our model training data does not sample the full temperature covariate space for minimum temperatures (Supplementary Fig. 1k), other research has suggested that climate may not be a significant predictor of C stocks³⁴, as increased production and decomposition may balance out at higher temperatures”

The environmental covariates used were expertly chosen and have been commonly used and tested in SOC variability studies. While using these variables within the cross-validation portion of the model makes sense, the use of these parameters does not show any particularly novel results globally. Many of these parameters are just general proxies and include multiple different ecosystem variables. I do think that the authors missed an opportunity with this large of a dataset to show potential insight into global trends of SOC, but also potential trends in those regions that are currently lacking data.

The impetus of the research was to develop a model of the spatial distribution of carbon stocks based on established drivers identified using expert opinion and discussion, and supported by the published literature. Our results do contain a section on the drivers of soil organic carbon in tidal marshes, where we have added additional text to discuss the role of temperature in determining tidal marsh carbon stocks (see above).

To clarify how well the training data samples the covariate space and to identify where certain environmental conditions are underrepresented, we have added a new visualisation (Fig. S1) and explain the approach in the methods.

“We visualised how well our training data captured the variability of the environmental covariates across the global tidal marsh extent, and as such, how potentially biased the environmental covariate data that was used to train the model was. To do this we sampled the covariate values for 10,000 points drawn randomly from across the global tidal marsh extent and compared these to the covariate values for the training locations (Fig. S1).”

In addition, the authors describe that currently the researchers have to rely on averages that are heavily biased towards temperate regions or not ecosystem specific but given the lack of data in other regions such as the tropics and the training data heavily in temperate regions, I'm not sure why this estimation is less biased. Did the authors test the bias? There needs to be a more extensive bias analysis for the authors to reliably say this study is less temperately biased than the others.

We have altered some of the language associated with this statement

“Without this information, the scientific community and practitioners have to rely on global averages that are not ecosystem-specific¹⁷ and that are based on data mainly from temperate regions¹⁸, or they must collect resource-intensive in-situ field measurements.”

However, we believe that highlighting the use of non ecosystem-specific data (ref 17) to quantify coastal wetland SOC is reasonable given that the dataset's FAQs highlights a lack of training data from coastal areas.

As raised above, we plot the distribution of the environmental covariates captured by the training data and compare it to the environmental covariate space from across the global tidal marsh distribution, to visualise how biased our training data is.

We have changed the wording of the start of the results to highlight why we think our study provides an improvement to current data.

“This estimate incorporates the spatial variability in tidal marsh SOC more adequately than previous studies, given that the model used training data from 3,710 unique locations^{19,20} and hypothesis-driven landscape-level drivers (Table S1), while previous estimates have relied on averaged values from a smaller subset of data.”

We also clearly set out how our results compare to previous studies.

“Previous global tidal marsh C stock estimates have taken a wide range of values. With lower values such as the 0.43 ± 0.03 Pg C estimated in the top 0.5 m from a dataset based mostly on North American tidal marshes¹⁸, continental SOC averages to 1 m multiplied by extent estimates (1.41–2.44 Pg)²⁵, or ranging between 0.86 and 1.35 Pg C¹⁶ estimated to a depth of 1 m from the SoilGrids map, a global machine-learning map from agricultural soils and terrestrial ecosystems data¹⁷. Conversely, simple calculations based on an average SOC value, applied to an overestimated tidal marsh extent have indicated that the global stock could be as high as 6.5 Pg C²². Our prediction of total global SOC in tidal marshes is significantly lower than this upper estimate, with our model predicting a range of 0.87-1.62 Pg C”

Individual Lines:

Line 85: Need to define SOC within the Main, not just the abstract.

Soil organic carbon now defined on the first line of the main

Line 105-108: The authors need more support to be able to say this. See temperature example in summary.

We've reworded the sentence and added justification to support the statement

“This estimate incorporates the spatial variability in tidal marsh SOC more adequately than previous studies, given that the model used training data from 3,710 unique locations^{19,20} and hypothesis-driven landscape-level drivers (Table S1), while previous estimates have relied on averaged values from a smaller subset of data”

Fig 3: Are any of these averages significantly different from one another? All the regions look very similar. It is strange that the Arctic has the highest average SOC.

Given the variation in the amount of post-AOA data for the different regions, we did not explicitly test whether there were significant differences between regions. However, per region statistics, including expected errors, are presented in Table S3.

Whilst the Arctic has the highest regional average value, in the text we highlight the challenges with interpreting this result.

“The large SOC predicted to 1 m in higher latitudes is influenced by limited training data and a low proportion of our predictions in the area of applicability (Fig. 3). In addition, the limited understanding of processes such as glacial isostatic adjustment and the impacts of relative sea level rise and how they influence C accumulation, decomposition and storage may profoundly alter estimates for the region, and remains an ongoing area of research.”

And in the Locations for priority sampling section

“For example, our analysis predicts high SOC across the high Arctic; however, this region is characterised by limited training data and thus high per pixel expected error. In our final analysis these areas are outside the AOA, and therefore there are significant uncertainties when estimating SOC in this region.”

In the initial version we had switched the figure legend for the x and y axes, this has now been corrected.

Line 185: How are the whiskers describing the error for the region, yet all of the whiskers show the same length? These don't seem to match up with the potential for value error shown in Fig 5, S3, and S4.

The whiskers in Figure 3 show the average expected model error for each realm, as such it may seem like they are similar, but they are not all the same length. We have clarified the difference between the regional error statistics and the finer scale maps in the text, and directed the reader to Table S3.

“The average expected error associated with our predictions was reasonably consistent at the regional level (Table S3, 0-30 cm layer: 43.0 - 52.5; 30-100 cm layer: 102.6 - 122.1); however, greater variation was more apparent at finer spatial scales (Fig. S3, Fig. S4).”

Line 203-205: How was the importance of these variables tested? Was it only in the cross-validation? Was no collinearity tested? The authors need to describe the methodology in much

more detail. Environmental variables are notoriously collinear, yet these lines imply that the authors did not test for variable interactions.

Cross validation is used to test the predictive performance of the model, we have clarified this in the methods.

“We used resampling-based cross-validation to provide an estimate of the predictive performance of the random forest model.”

A variable importance methodology is used to create Figure 4, we have added a section to the methods to describe it.

“Within the final random forest, variable importance was set to “impurity” within the ranger package, corresponding to the Gini index for classification. This was used to identify the relative importance of the environmental covariates to the SOC predictions.”

There was little collinearity between the continuous variables. We have text to that effect and a new figure (Fig. S7)

“Collinearity between the continuous variables was visualised using the corrplot package (version 0.92), and identified generally low correlations between the variables (Fig. S7), and below the threshold of $|r| > 0.7$.”

Line 255: How much lower for tropics compared to global? The model does not show that much of a decrease between the averages or ranges.

We added the statistics from Table S3 to clarify this statement

“Our analysis suggests that the average SOC per unit area is lower (Table S3, 0-30 cm layer: 5 - 23%; 30-100 cm layer: 9 - 29%) for three out of the five tropical regions compared to the global average; a result supported by the currently available studies⁴⁸.”

Lines 488: How different is too different? From Figures S1 and S2, it appears that a significant portion of the tropics and Arctic are not included in the AOA. With so much error, what can truly be determined from these estimations? Besides the fact that these areas need more ground-truthing.

We have clarified the AOA approach in the methodology.

“To ensure our predictions were bounded with the environmental envelope of our training data, we implemented the area of applicability (AOA) methodology, introduced by Meyer and Pebesma 2021²⁴, to mask out areas where the model was not able to learn about the relationship between the predictors and the response (here, SOC density). We specifically excluded areas with a different covariate space where predictions of carbon stocks would be uncertain because of a lack of training and validation data. The threshold for determining the AOA was based on the outlier-removed maximum DI of the training data, i.e. data larger than the 75th percentile plus 1.5 times the interquartile range of the DI values of the cross-validated training data.”

It is true that a significant portion of the model predictions for the Arctic and tropics are removed using the AOA approach, therefore the estimations that are presented in paper are those where the error associated with the prediction is less. We have highlighted the significance of applying the AOA in the first paragraph of the results.

References:

Why do the references suddenly go in alphabetical order halfway through? This formatting may need to be checked.

This is because references 54-256 relate to the training data that was used to parametrise the random forest model and are cited at the same point in the text and thus displayed in alphabetical order

Supplemental:

Table S1: Are these variables decided by literature or by expert opinion/discussion (Lines 331-334)? Otherwise, very helpful and informative table.

Table legend clarified as to how the variables were selected.

“These variables were selected using expert opinion and discussion, along with previous studies investigating the variables identified for their associations with SOC in vegetated coastal ecosystems³⁻⁵, and supported by evidence from the published literature.”

Table S2: Also, good table.

Thank you

Figure S4: The color choices for plot a and c make it difficult to see any variation in the maps. Given these are the key error maps for the 30-100 meter, these should be as easy to see as Figure S3.

We have redrawn Figure S3 and S4 so that the error classes are different between panels a) and c), and noted this difference in the figure legends

“NB. please note the difference in error classes between panels a) and c).”

Reviewer #2 (Remarks to the Author):

General Comments:

The manuscript presents a compelling methodology for quantifying the current carbon stocks in coastal wetlands globally. The methodology utilises a broad literature base for measured SOC at specific locations, a new global tidal marsh extent map and a machine-learning approach including environmental covariates (identified by a broad input from co-authors) as potential drivers of soil carbon density. The manuscript does not, however, attempt to discuss discrepancies with other methodologies for determining SOC stocks, for example, the manuscript describes the estimate for China to be 19 Tg, while a recent paper in GCB (Xia et al) estimates the stocks to exceed 50 Tg.

We have included some text comparing our results to the work of Xia et al., 2022, and highlight the key drivers in differences between different national level predictions.

“Not all our findings are so well aligned with other studies. For example, we predict 19.3 Tg for China, while 57 Tg C was estimated in an earlier study (although this also included the contribution of mangroves and tidal flats)²⁹. Such differences are likely to be driven by several factors, most strongly of which is the area of tidal marsh estimated for each country²⁶. However, the availability of training data that accurately captures the variability of environmental conditions and the inclusion of finer scale model predictors (e.g., data on tidal marsh plant communities) of C stocks will impact estimates.

Table 4 describes the key drivers for SOC stocks using random forest methodology, however, it is not entirely clear what the NDVI was reporting- was this used as a proxy for vegetation density or vegetation type or both? Perhaps this needs clarification in the methods.

More detail added in Table S1.

“Indices such as NDVI are capable of discriminating between broad tidal marsh vegetation communities.”

And in the text describing the variable.

“We used the Normalised Difference Vegetation Index (NDVI) as a proxy for distinguishing vegetation type and the source of SOC. NDVI has been shown to be able to discriminate between broad classes of tidal marsh vegetation.”

The manuscript provides valuable insights into regions that require empirical data to further train the models. In particular, the Arctic regions and tropical regions require further attention. It would be interesting to further understand why mangrove forests in the tropics give high variability in predictions.

We have added text to the ‘locations for priority sampling’ section to address this point.

“Tidal marshes in tropical regions are an important component of the coastal seascape, yet our understanding of both their extent and SOC stocks is limited by available data. Within our model there was variation in the predicted SOC per unit area across tropical realms (Table S3), but the majority of predictions were associated with high expected error (Fig. 5). The mechanisms behind the variability in SOC are unknown but may relate to variation in the community composition and productivity of tropical tidal marshes.”

The outlook and policy implication section is also compelling, based on data presented in the manuscript and a sound understanding of ecosystem restoration.

We thank the reviewer for the positive comment.

The model is trained with a robust range of environmental covariates. However, anthropogenic disturbance is not factored into model outputs, while it is generally well understood that degradation via erosion (as an example) can also severely impact SOC stocks. Perhaps these areas were excluded from the current assessment, but this may need some further explanation.

In the 'drivers of soil organic carbon in tidal marshes' section we have added some text to describe how anthropogenic drivers may alter SOC stocks compared to our model predictions.

"Finally, our analysis provides a static estimate of tidal marsh SOC stocks driven by environmental covariates. However, it is well established that natural and anthropogenic drivers can impact tidal marsh persistence and condition, and as such their SOC stocks⁸. Anthropogenic disturbances could both deplete C storage (e.g. from erosion or direct habitat removal, although such impacts would remove them from our map and model) or increase C storage (e.g., from improved productivity due to nutrient additions) beyond what would be predicted by our model."

The manuscript is generally very well written and highly polished. There could perhaps be improved consistency in the use of abbreviations/ chemical symbols, in particular carbon and C are used interchangeably, and should be consolidated to C. Otherwise, the manuscript is well suited for Nature Communications and should achieve significant utilization by a broad audience.

We thank the reviewer for positive comments on the writing of the manuscript. For consistency we have ensured the Carbon is consolidated to C where appropriate, and Blue Carbon Ecosystems is BCEs.

Specific comments:

54 "used training data from 3,710 unique locations" rather, the model is trained on a globally distributed empirical dataset. This minor amendment may improve the readability for a non-expert in modelling approaches.

Apologies if we misunderstood this comment, we have changed the sentence in the main to:

"This estimate incorporates the spatial variability in tidal marsh SOC more adequately than previous studies, given that the model used training data from 3,710 unique locations^{19,20} and hypothesis-driven landscape-level drivers (Table S1), while previous estimates have relied on averaged values from a smaller subset of data."

67 SOC

Acronym added to sentence

68 waterlogged and temporarily waterlogged soils?

Sentence edited to

“temporarily or permanently waterlogged soils”

70-72 The concept of C saturation in terrestrial systems is driven by edaphic controls (mainly), while this is less relevant in wetlands- where protection of C relies on protection of POM and MAOM by lower microbial oxidation. The increase in level of wetlands will rely heavily on allochthonous inputs to bring in the mineral matter. I think this sentence could use a little more explanation.

We've rephrased the sentences as follows

“Tidal marsh soils are capable of accreting vertically with sea level rise with inputs from allochthonous and autochthonous sources, thus limitations to C accumulation are far less likely to occur in marshes compared to terrestrial ecosystems, providing potential for continuous climate change mitigation benefits.”

91 in-situ in italics?

In-situ italicised in text

118-120 The high variability in tropical regions- particularly mangrove forests, may also be underestimated. Perhaps this should be mentioned here.

We have added a sentence to highlight the potential for underestimated values in the tropics

“In addition, the combination of high expected error of predictions resulted in many areas in the tropics being removed from the statistics (Fig S3, Fig S4). These removals, coupled with our currently incomplete understanding of the full distribution of tidal marshes⁴, suggests that carbon stocks could also be underestimated in the tropics.”

323 not sure that SOC needs to be redefined here

Changed to SOC

366 Total does not require capitalization.

Capital letter removed

491 SOC

Changed to SOC, and document checked for consistency.